

# The application of sudoscan for screening microvascular complications in patients with type 2 diabetes

Kun Lin[1], Yixi Wu[1], Shuo Liu[2], Jiaqi Huang[1], Guishan Chen[1] and Qiong Zeng[2]

[1] Department of Endocrinology, The First Affiliated Hospital of Shantou University Medical College, Shantou, China
[2] Department of Neurology, The First Affiliated Hospital of Shantou University Medical College, Shantou, China

## ABSTRACT

The aim of the study was to evaluate the performance of sudoscan in screening diabetic microvascular complications in patients with type 2 diabete mellitus (T2DM). 515 patients with T2DM aged from 23 to 89 years were included for analysis in our study. The mean age was $60.00 \pm 11.37$ years and the mean duration of T2DM was $8.44 \pm 7.56$ years. Electrochemical skin conductance (ESC) in hands and feet was evaluated by SUDOCAN. Diabetic peripheral neuropathy (DPN) was diagnosed in 378 patients (44.3%), diabetic kidney disease (DKD) in 161 patients (31.26%), diabetic retinopathy (DR) in 148 patients (28.74%). Hands and feet ESC was significantly and independently associated with the presence of DPN, DKD and DR. Patients with a lower ESC ($<60$ µS) had 5.63-fold increased likelihood of having DPN, 4.90-fold increased likelihood of having DKD, 1.01-fold increased likelihood of having DR, than those with a higher ESC. Age, duration of T2DM, smoking, renal function and vibration perception thresholds were negatively correlated with ESC. Sudoscan parameters were correlated with diabetic microvascular complications, especially with DPN. Sudoscan could be an effective screening tool in primary health care for early screening microvascular complications.

## INTRODUCTION

Diabetic microvascular complications mainly including diabetic peripheral neuropathy (DPN), diabetic kidney disease (DKD) and diabetic retinopathy (DR) are the major cause of disability and death. Early identification and diagnosis of diabetic complications has a great clinical value in reducing or delaying the occurrence and development of diabetic chronic complications (*Zimmet, Alberti & Shaw, 2001*).

There are many screening methods for diabetic microvascular complications. For example, 10g Monofilament & Tuning Fork (128 Hz), electromyogram, vibration perception thresholds test (VPT), Michigan neuropathy screening instrument (MNSI) can be used to screen DPN. Estimated glomerular filtration rate (EGFR) and urinary albumin/creatinine ratio (UACR) are the main methods of screening DKD. However, all these screening methods have their own shortcomings. In addition, the efficiency of

Corresponding author
Qiong Zeng, jennyzengch@126.com

different examinations in the diagnosis of diabetic complications is quite different. It is of great significance to find a simple screening method which can be popularized in grass-roots communities and can predict the risk of multiple complications early.

Although the pathogenesis of diabetic microvascular complications is complex, different complications have a common pathological basis (*Vinik et al., 2003*; *Sytze Van Dam et al., 2013*). A neurovascular concept of diabetic complications has been postulated (*Sytze Van Dam et al., 2013*). In recent years, as a new detection method of diabetic complications, sudoscan has been initially applied in clinical practice (*Handelsman et al., 2015*). As we know, sweat glands are innervated by small unmyelinated sympathetic C-fibers, and these small C-fibers can be affected early in the neuropathic process (it even occurs during the prediabetic stage) (*Müller et al., 2013*). Sudoscan is a device to test the ability of palmar and plantar sweat glands releasing chloride ions under the electrochemical activation by using electrochemical principle, and then to measure the sweat function. According to the status of sweat secretion function at the extremities, sudoscan output the results of electrochemical skin conductance (ESC), to determine whether there are diabetic complications (*Vinik, Nevoret & Casellini, 2015*). At present there is a lack of comparative study on the diagnostic value of sudoscan for different diabetic complications in type 2 diabete mellitus (T2DM).

The aim of this study was to analyze the relationship between the Sudoscan parameters and the diabetic microvascular complications in T2DM, and to evaluate the clinical value of sudoscan in diagnosis of microvascular complications in T2DM in China.

## MATERIALS & METHODS

### Subjects

All patients included in the study were recruited between November 2020 to July 2021 from the department of Endocrinology, the First Affiliated Hospital of Shantou University Medical College. A total of 515 patients (male 274 and female 241) with T2DM aged from 23 to 89 years were finally included. The exclusion criteria were as follows: other types of diabetes, pregnancy, mental and neurological disorders, with acute complications of diabetes mellitus, inflammation, cancer, severe liver or kidney dysfunction. The study was fully approved by the ethics committee of the First Affiliated Hospital of Shantou University Medical College (Approval Document Number: B-2020-194). All patients gave written informed consent to participate in the study. All procedures conformed to the tenets of the Declaration of Helsinki.

### SUDOSCAN+ device

The evaluation of sudomotor function was measured with the Sudoscan medical device (Impeto Medical, France), consisting of a set of two electrodes for feet and hands connected to a computer. The patients placed their palms and soles on electrodes for about 3 min. Quantitative results were expressed as electrochemical skin conductance (HESC and FESC, μS), asymmetry ratio value (HASYM and FASYM, %) in hands and feet. In addition, SUDOSCAN had built-in algorithms which integrate electrochemical skin conductance with age to produce a score that estimates current risks of DKD (sudoscan modification of diabetic renal disease, SUDOSCAN-MDRD) (*Ozaki et al., 2011*). In the present study

the patients were divided into ESC normal group and ESC abnormal group according to ESC results. Patients with HESC and FESC ≥ 60 µS were included in the ESC normal group, while patients with HESC or FESC <60 µS were included in the ESC abnormal group (*Vinik, Nevoret & Casellini, 2015*).

## Laboratory tests

Blood urea nitrogen (BUN), serum creatinine (Cr), uric acid (UA), fasting C-peptide(F-CP), 2 h Postprandial C-peptide(2hP-CP), Glycosylated hemoglobin (HbA1c) and serum lipids including total cholesterol (TC), triglyceride (TG), high-density lipoprotein cholesterol (HDL-C) and low-density lipoprotein cholesterol (LDL-C) were assessed by an automatic biochemical analyzer (COULTER LX20; BECKMAN, USA).

## Assessment of microvascular complications

The diagnostic criteria for DPN were based on the 2010 American Diabetes Association (ADA) DPN clinical diagnostic criteria (*Tesfaye et al., 2010*). Physical examinations of DPN included temperature sensation, 10 g Monofilament, Tuning Fork (128 Hz), vibration perception thresholds test, ankle reflex and acupuncture pain test. Symptoms were assessed on the basis of information collected during medical history. DKD was defined as the presence of UACR >30 µg/mg or eGFR of less than 60 ml/min/1.73 m2 (or both) (*National Kidney Foundation, 2002*). EGFR was calculated by Cockcroft-Gault formula (*Thompson-Martin, McCullough & Agrawal, 2015*). DR was examined by fundus photography (KOWA, Japan). Carotid intima and extremity vessels were examined by Color Doppler (Siemens, Germany).

## Statistical analyses

Statistical analyses were computed using SPSS 19.0 (SPSS Inc., Chicago, IL, USA) and medcalc19.0 (MedCalc Software bvba, Ostend, Belgium). The measurement data were presented as the mean ± standard deviation, and the numeration data were expressed as ratio or constituent ratio. Differences for continuous variables and categorical variables among groups were assessed by ANOVA test and $\chi 2$ test, respectively. Pearson correlation method was used to determine correlation between ESC and clinical variables. Forward conditional binary logistic regression analysis was used to find the independent risk factors for DPN, DKD and DR. Taking DPN as the dependent variable, the logistic regression model included the following variables: sex, age, duration, hypertension, smoking, HbA1c, LDL, VPT, ESC normal or abnormal. Taking DKD as the dependent variable, the model included: sex, age, duration, hypertension, smoking, HbA1c, LDL, eGFR, UACR, ESC normal or abnormal. Taking DR as the dependent variable, the model included: sex, age, duration, hypertension, smoking, HbA1c, LDL, eGFR, UACR, ESC normal or abnormal. Receiver operating characteristics (ROC) curve analysis were used to calculate sensitivity and specificity and to determine the diagnostic accuracy of the tests. The area under the curve (AUC) were compared by $Z$-test.

## RESULTS

### Basic clinical characteristics: more DPN, DKD and DR in ESC normal group

The study participants consisted of 515 individuals of T2DM aged from 23 to 89 years. All are Han Chinese. The mean age was $60.00 \pm 11.37$ years and the mean duration of T2DM was $8.44 \pm 7.56$ years. DPN was diagnosed in 378 patients (44.3%), DKD in 161 patients (31.26%) and DR in 148 patients (28.74%). The mean HESC and FESC were $55.63 \pm 18.44\,\mu S$ and $61.76 \pm 20.51\,\mu S$, respectively. According to the ESC value, the patients were divided into ESC abnormal group and ESC normal group. Baseline characteristics of both groups were presented in Table 1. There were significant differences in age ($59.78 \pm 11.38$ vs $57.64 \pm 11.25$, $P = 0.040$), presence of coronary heart disease (4.59% vs 2.64%, $P = 0.027$), presence of DPN (88.99% vs 42.86%, $P < 0.001$), presence of DKD (39.76% vs 16.40%, $P < 0.001$) and presence of DR (30.58% vs 25.53%, $P = 0.012$) between both groups.

### Correlation analyses: ESC was independent related factor of microvascular complications

Correlation analyses among parameters with HESC and FESC were shown in Table 2. Both HESC and FESC were significantly and negatively correlated with age, T2DM duration, UACR, Cr and VPT. EGFR were positively associated with HESC and FESC. Furthermore, there was a significant correlation between FESC and sex. FESC in male individuals was significantly lower than that in female ($59.53 \pm 21.66$ *vs* $64.28 \pm 18.89$, $P = 0.008$).

To further clarify the independent related factors of microvascular complications, A binary logistic regression analysis (forward conditional) was used to determine factors associated with diabetic microvascular complications, (Table 3). The results showed that the significant factors for DPN included ESC ($P < 0.001$, OR = 5.631) and VPT ($P < 0.001$, OR = 1.329); the significant factors for DKD included ESC ($P < 0.001$, OR = 4.895), UACR ($P < 0.001$, OR = 1.025) and eGFR ($P = 0.037$, OR = 0.987); the significant factors for DR included eGFR ($P < 0.001$, OR = 0.984), duration ($P = 0.016$, OR = 1.037) and ESC ($P = 0.035$, OR = 1.014). After confirming the correlation between ESC and complications, the diagnostic value of sudoscan parameters for the diagnosis of microvascular complications were further explored by ROC curve analysis (Table 4 and Fig. 1).

## DISCUSSION

In this large sample of Chinese inpatients with type 2 diabetes, we analyzed the discriminative value for microvascular complications of sudoscan parameters. We found that (i) hands and feet ESC was significantly and independently associated with the prevalence of DPN, DKD and DR. (ii) Patients with a lower ESC (<60 μS) had 5.63-fold increased likelihood of having DPN, 4.90-fold increased likelihood of having DKD, 1.01-fold increased likelihood of having DR, than those with a higher ESC. (iii) Age, duration of T2DM, smoking, renal function and VPT were negatively correlated with ESC; Blood

**Table 1  Subject demographics and clinical characteristics.**

| Characteristics | ESC normal ($n = 188$) | ESC abnormal ($n = 327$) | $P$-value |
|---|---|---|---|
| Age (years) | $57.64 \pm 11.25$ | $59.78 \pm 11.38$ | 0.040 |
| Male | 92 | 182 | 0.144 |
| Female | 96 | 145 | |
| Duration of DM (years) | $7.73 \pm 6.78$ | $8.85 \pm 7.95$ | 0.104 |
| BMI (kg/ m$^2$) | $23.47 \pm 3.59$ | $22.99 \pm 3.46$ | 0.136 |
| SBP (mmHg) | $137.71 \pm 18.48$ | $138.07 \pm 21.41$ | 0.846 |
| DBP (mmHg) | $85.74 \pm 11.03$ | $85.06 \pm 12.76$ | 0.543 |
| F-CP | $0.38 \pm 0.24$ | $0.40 \pm 0.37$ | 0.406 |
| 2hP-CP | $0.82 \pm 0.65$ | $0.80 \pm 0.73$ | 0.808 |
| Smoking | 17.46% | 25.38% | <0.001 |
| Drinking | 7.41% | 7.03% | 0.752 |
| Coronary heart disease | 2.64% | 4.59% | 0.027 |
| Cerebral infarction | 14.29% | 13.15% | 0.471 |
| CIMT (mm) | $0.96 \pm 0.18$ | $0.99 \pm 0.19$ | 0.078 |
| VPT (V) | $13.79 \pm 6.64$ | $17.23 \pm 9.52$ | <0.001 |
| HbA1c (%) | $9.98 \pm 2.38$ | $10.08 \pm 2.62$ | 0.670 |
| UACR | $92.04 \pm 355.53$ | $368.80 \pm 1264.38$ | <0.001 |
| Cr ($\mu$mol/l) | $87.00 \pm 49.66$ | $97.45 \pm 58.02$ | 0.046 |
| eGFR (ml/min/1.73 m$^2$) | $78.33 \pm 29.41$ | $70.12 \pm 31.02$ | 0.004 |
| UA ($\mu$mol/l) | $346.38 \pm 102.91$ | $356.52 \pm 113.35$ | 0.319 |
| TC (mmol/l) | $5.26 \pm 1.58$ | $5.23 \pm 1.63$ | 0.799 |
| TG (mmol/l) | $2.23 \pm 2.11$ | $2.08 \pm 2.09$ | 0.456 |
| HDL (mmol/l) | $1.19 \pm 0.44$ | $1.12 \pm 0.39$ | 0.086 |
| LDL (mmol/l) | $3.29 \pm 0.96$ | $3.28 \pm 1.02$ | 0.931 |
| DPN | 42.86% | 88.99% | <0.001 |
| DKD | 16.40% | 39.76% | <0.001 |
| DR | 25.53% | 30.58% | 0.012 |

**Notes.**

SBP, systolic blood pressure; DBP, diastolic blood pressure; HbA1c, glycosylated hemoglobin; Cr, serum creatinine; UA, uric acid; TC, total cholesterol; TG, triglyceride; HDL, high-density lipoprotein cholesterol; LDL, low-density lipoprotein cholesterol; CIMT, carotid intima-media thickness.

pressure, HbA1c and $\beta$ cell function were not correlated with sudoscan parameters. (iv) The cut-off point of ESC in the diagnosis of microvascular complications was about 60 $\mu$S. The AUC of HESC/FESC to identify DPN, DKD and DR were all above 62%.

Traditional DPN screening tools, such as 10 g Monofilament & Tuning Fork, are subjective and non quantitative measurement. Neuroelectromyography is an invasive procedure and reflects the injury of large myelinated fibers. However, the unmyelinated, thin type C fibers of the sympathetic nervous system are usually neglected because of the limited evaluation methods. Sudoscan can remedy the above shortcomings (*Ziemssen & Siepmann, 2019*). Previous clinical studies have indicated the value of sudoscan in DPN. ESC was significantly negative associated with Neuropathy Disability Score, Neurological Symptom Score, MNSI and VPT in diabetes; the sensitivity and specificity of ESC in diagnosis of DPN is between 73–87.5 and 55–76.5% respectively, the cutoff value was
**Table 2  Correlation analysis between HESC/FESC and clinical variables.**

| Variables | HESC | | FESC | |
|---|---|---|---|---|
| | r | P-value | r | P-value |
| Age | −0.156 | <0.001 | −0.104 | 0.019 |
| Sex | 0.078 | 0.077 | 0.116 | 0.009 |
| Duration of DM | −0.098 | 0.026 | −0.112 | 0.011 |
| SBP | −0.013 | 0.762 | −0.046 | 0.300 |
| DBP | 0.012 | 0.792 | 0.012 | 0.788 |
| Smoking | −0.134 | 0.002 | −0.095 | 0.031 |
| HbA1c(%) | −0.056 | 0.210 | −0.002 | 0.956 |
| F-CP | 0.029 | 0.516 | −0.054 | 0.226 |
| UACR | −0.164 | <0.001 | −0.120 | 0.009 |
| Cr | −0.165 | <0.001 | −0.156 | <0.001 |
| eGFR | 0.208 | <0.001 | 0.157 | <0.001 |
| CIMT | −0.082 | 0.086 | −0.082 | 0.088 |
| VPT | −0.245 | <0.001 | −0.366 | <0.001 |

**Table 3  Risk factors for microvascular complications in binary logistic regression (forward conditional).**

| | | B | S.E. | Wald | P value | OR | 95% CI for OR | |
|---|---|---|---|---|---|---|---|---|
| | | | | | | | Lower | Upper |
| DPN | ESC | 1.728 | 0.330 | 27.392 | <0.001 | 5.631 | 2.948 | 10.757 |
| | VPT | 0.285 | 0.045 | 39.526 | <0.001 | 1.329 | 1.216 | 1.453 |
| DKD | ESC | 1.588 | 0.371 | 18.370 | <0.001 | 4.895 | 2.368 | 10.121 |
| | UACR | 0.025 | 0.003 | 55.615 | <0.001 | 1.025 | 1.019 | 1.032 |
| | eGFR | −0.012 | 0.006 | 4.332 | 0.036 | 0.987 | 0.976 | 0.999 |
| DR | eGFR | −0.016 | 0.004 | 14.196 | <0.001 | 0.984 | 0.976 | 0.992 |
| | Duration | 0.036 | 0.015 | 5.817 | 0.016 | 1.037 | 1.007 | 1.068 |
| | ESC | −0.013 | 0.006 | 5.202 | 0.035 | 1.014 | 1.003 | 1.026 |

52–61 μS, from different studies (*Casellini et al., 2013*; *Yajnik et al., 2012*; *Eranki et al., 2013*; *Selvarajah et al., 2015*; *Mao et al., 2017*). In our study, The AUC of FESC to evaluate DPN were 0.71, the cut-off value was 61 μS, the sensitivity and specificity was 79.0% and 75%, respectively; ESC was negatively correlated with VPT; ESC were independently associated with DPN. The results were basically consistent with the above studies. The results suggested that sudoscan can be used as a clinical screening device for diabetic neuropathy. The previous studies and our study all indicated that the sensitivity of ESC for diagnosing DPN is higher than the specificity. An interpretation is that the small unmyelinated sympathetic C-fibers is affected earlier than the large myelinated nerve fiber in the course of diabetes (*Müller et al., 2013*).

DPN and DKD have similar pathological progress and parallel in occurrence and development (*Tahrani et al., 2014*). Therefore, sudoscan may also reflects the risk of DKD. *Freedman et al. (2014)* found a significant correlation between ESC and eGFR in

**Table 4 Comparison of diagnostic value among different parameter for microvascular complications.**

|  | Parameter | AUC | *P*-value | cut-off point | Sensitivity (%) | Specificity (%) | Youden index |
|---|---|---|---|---|---|---|---|
|  | HESC | 0.71 | <0.001 | 73 μS | 61 | 71 | 0.32 |
| DPN | FESC | 0.71 | <0.001 | 61 μS | 79 | 65 | 0.44 |
|  | VPT | 0.81 | <0.001 | 11 V | 87 | 71 | 0.58 |
|  | HESC | 0.64 | <0.001 | 59 μS | 57 | 67 | 0.24 |
|  | FESC | 0.65 | <0.001 | 59 μS | 73 | 50 | 0.24 |
| DKD | SUDOSCAN-MDRD | 0.65 | <0.001 | 58 | 54 | 71 | 0.25 |
|  | UACR | 0.90 | <0.001 | 30 mg/g | 88 | 92 | 0.80 |
|  | eGFR | 0.71 | <0.001 | 60 ml/min/1.73 m$^2$ | 79 | 54 | 0.33 |
|  | HESC | 0.64 | <0.001 | 61 μS | 57 | 73 | 0.30 |
| DR | FESC | 0.62 | <0.001 | 70 μS | 54 | 69 | 0.23 |
|  | eGFR | 0.59 | <0.001 | 78 ml/min/1.73 m$^2$ | 44 | 72 | 0.15 |

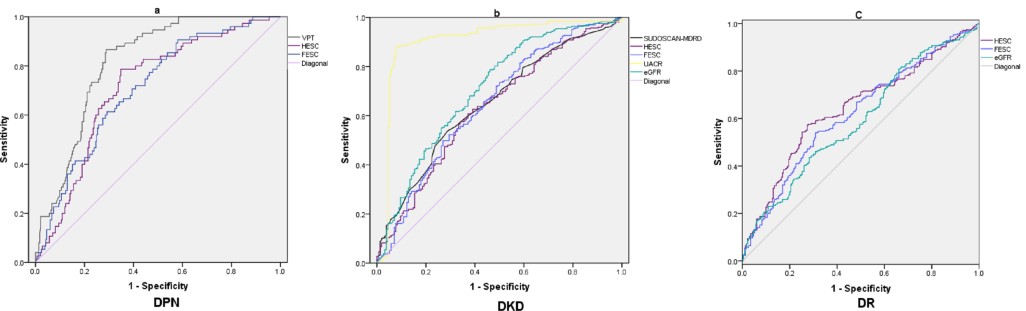

**Figure 1 ROC curves of different parameter in screening microvascular complications.** (A) ROC curves of FESC, HESC and VPT to diagnose DPN; (B) ROC curves of SUDOSCAN-MDRD, FESC, HESC, UACR and eGFR to diagnose DKD; (C) ROC curves of FESC, HESC and eGFR to diagnose DR.

African American and European American patients with T2DM, suggesting that ESC measured by sudoscan can be used to predict the risk of DKD in clinical practice. *Luk et al. (2015)* and *Xue, De-ming & Lin-tao (2016)* found that the cut-off value of SUDOSCAN-MDRD to evaluate DKD was 53 and 58, the sensitivity and specificity was 72–54% and 68–71%, respectively. In the present study, The AUC of SUDOSCAN-MDRD to evaluate DKD were 0.65, the cut-off value was 58, the sensitivity and specificity was 54% and 71%, respectively. Furthremore the diagnostic value of SUDOSCAN-MDRD and FESC were slightly lower than eGFR ($Z = 2.612$, $P = 0.009$ and $Z = 1.759$, $P = 0.078$, respectively) and far lower than UACR($Z = 11.86$, $P < 0.001$ and $Z = 11.09$, $P < 0.001$, respectively). This may be related to the characteristics of diabetic nephropathy in China. Microalbuminuria is the main manifestation of early diabetic nephropathy in China (*Wan, Xu & Dong, 2015*). It suggested that SUDOSCAN-MDRD and ESC can not replace eGFR and UACR in Chinese patients with DKD.

To our knowledge, few study evaluated the clinical value of ESC in diagnosis of DR in T2DM. The pathogenesis in DR is similar to other microvascular complications. The downstream processes of persistent hyperglycemia, including the activation of protein

kinase C, the activation of polyol pathway, and the formation of advanced glycation end products, are considered as the causes of diabetic microvascular changes and direct nerve injury (*Vinik et al., 2003*). *Wang et al. (2017)* explored the relationship between autonomic nerve dysfunction-assessed by ESC and ocular abnormality in T2DM. The result showed that hands and feet ESC were positively associated with lens (OR = 1.055, $P < 0.001$) and vitreous (OR = 1.044, $P < 0.01$) abnormality. In the present study, eGFR and ESC were independently associated with DR. The AUC of FESC to diagnose DR were 0.62; the diagnostic value of FESC for DR did not differ significantly from that of eGFR ($Z = 0.907$, $P = 0.364$).

The study dedicated that duration of T2DM, age and smoking were negatively correlated with the sudoscan parameters. This is consistent with previous studies (*Tesfaye et al., 1996*). The older the age, the longer the course of diabetes, and the worse the sweating function of hands and feet, which indicating the more serious the damage of nerves innervating sweat glands. This further supported that patients of DM may have different degrees of nerve damage in the early stage (*Tesfaye, Chaturvedi & Eaton, 2005*; *Pang et al., 2008*). This study also found that FESC in male was significantly lower than that in female. In previous clinical studies, it was also found that the diagnosis of diabetic neuropathy in men was earlier than that in women (*Kamenov, Parapunova & Georgieva, 2010*).

In the present study, the incidence rate of coronary heart disease in ESC abnormal group was significantly higher than that in ESC normal group. However, no significant correlation was found between ESC and diabetic macrovascular complications (CIMT, coronary heart disease, cerebral infarction, etc.). This may be related to the study design (non prospective study) and research focus (microvascular complications). One large prospective study had found that ESC was associated with cardiovascular events and death in patients with type 2 diabetes (*Lim et al., 2019*). The adjusted risk ratio of cardiovascular disease (CVD) in patients with low ESC was 3.11.

There are still several limitations in our study. First, this was a cross section study. Second, only type 2 diabetes inpatients were included. Third, the mean duration of diabetes was 7-8 years in the sample with mean HbA1c 10%. Is it possible that sudoscan is only applicable in these high risk diabetes patients? Further investigation is needed to explore sudoscan diagnostic performance in newly-onset diabetes or pre-diabetes patients.

## CONCLUSIONS

In conclusion, sudoscan was a noninvasive, rapid, simple and repeatable device. The results of ESC were stable and reliable. It will not be affected by the subjectivity of the operator and the environment. Based on this study, sudoscan had an effective diagnostic value for microvascular complications, especially for DPN, although it can not replace the classic diagnostic methods. In view of its convenience and advantages of providing multiple complications risk, sudoscan is worthy of promotion and application in primary medical institutions for early screening microvascular complications. In diabetes specialists in China, it may be not recommended to replace the classic diagnostic gold index and test.

## ACKNOWLEDGEMENTS

Acknowledgments were given to patient advisers, all staff and nurses who work at the department of Endocrinology, The First Affiliated Hospital of Shantou University Medical College, Shantou, China.

### Funding

The authors received no funding for this work.

### Competing Interests

The authors declare there are no competing interests.

### Author Contributions

- Kun Lin conceived and designed the experiments, prepared figures and/or tables, authored or reviewed drafts of the paper, and approved the final draft.
- Yixi Wu performed the experiments, prepared figures and/or tables, and approved the final draft.
- Shuo Liu analyzed the data, authored or reviewed drafts of the paper, and approved the final draft.
- Jiaqi Huang and Guishan Chen performed the experiments, authored or reviewed drafts of the paper, and approved the final draft.
- Qiong Zeng analyzed the data, authored or reviewed drafts of the paper, and approved the final draft.

### Human Ethics

The following information was supplied relating to ethical approvals (i.e., approving body and any reference numbers):

The First Affiliated Hospital of Shantou University Medical College granted Ethical approval to carry out the study within its facilities.

### Data Availability

Raw data are uploaded as Supplementary File.

### Supplemental Information

Supplemental information for this article can be found online at http://dx.doi.org/10.7717/peerj.13089#supplemental-information.

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
