# Peer review of "The application of sudoscan for screening microvascular complications in patients with type 2 diabetes"

_PeerJ, doi:10.7717/peerj.13089_

## Round 0.1 · original submission · Major Revisions

Although the paper is interesting and has merit, the authors should address all the concerns raised by the reviewers, regarding the experimental design and interpretation, especially in the light of statistical analysis.

·

Excellent Review

This review has been rated excellent by staff (in the top 15% of reviews)
EDITOR COMMENT
In my opinion, this review has many specific strengths. It is well organised and detailed, while comments are accurate, objective and constructive. My impression is that this review may really help the authors to improve the quality of their manuscript.

Basic reporting

See additional comments

Experimental design

See additional comments

Validity of the findings

See additional comments

Additional comments

The authors submitted a paper in which they evaluated the performance of the Electrochemical Skin Conductance, measured with a Sudoscan device, as a potential marker of different microvascular complications in type 2 diabetes patients, namely diabetic neuropathy (DPN), diabetic kidney disease (DKD) and diabetic retinopathy (DR).
Additionally, the authors analyzed a derived score given by the device (SUDOSCAN modification of diabetic renal disease) also as a potential marker of DKD.
Each condition was determined according to standard international accepted criteria.
Subjects were divided into 2 groups according to ESC results (normal > 60 µS; abnormal < 60 µS).
The authors reported significant correlations between hands and feet ESC and several clinical variables.
The authors then performed a binary logistic regression analysis and found that a mean of the ESC from the 4 limbs was a significant independent predictor for DPN and DR. The authors also reported that the score SUDOSCAN-MDRD was also a significant independent predictor for DKD.
Finally, the authors performed a ROC curve analysis of the diagnostic value of hands and feet ESC for DPN and DR as well as SUDOSCAN-MDRD for DKD.
The authors then concluded that ESC, as measured by the Sudoscan device, has an effective, but limited, value as a diagnostic tool for diabetic microvascular complications.

Despite not being the first study to investigate this relationship, the article is interesting and has some strong points in its favor, in particular, the large number of patients studied, with a wide range of ages.
However, there are some points that need clarification before a publication endorsement.
Concerns:
1- The article would benefit from a thorough revision regarding the language used.
2- Across the article, the terms Sudoscan and ESC are used interchangeable. This should be avoided. The technique used is the Electrochemical Skin Conductance. The device used for this is called Sudoscan. The name of the device should not be used as a technique. Please clarify this throughout the text
3- Clarification regarding SUDOSCAN-MDRD is needed. The authors do not explain this score or reference any publication in the Methods section. Why was this score used in the ROC analysis instead of the ESC values?
4- In the results section, the authors stated that FESC was lower in males when compared to females (59.53 ± 21.66 vs 64.28 ± 18.89, P = 0.002). However, when verifying this values with the raw data supplied, the p value obtained is not consistent with the one reported.

As can be seen in the picture above, the p value of 0.002 refers to the Levene Test For the Equality of Variances. The correct p value to be reported should be p = 0.008, since the Levene test suggests that the homogeneity of variances assumption is violated.
This is not an overly important mistake but should warn the authors to double check their results interpretation.

5- Regarding the binary logistic regression analysis, the results are somewhat confusing. What are the results of each model? Where all variables considered for the final models? Looking at Table 3, it appears that all variables were considered for the 3 models analyzed. If so, what is the clinical rationale to consider BMI, hypertension or eGFR for the DPN model? Or the clinical rationale to consider VPT for the DKD or DR models? This needs to be clarified.
6- Another point in the binary logistic regression is the choice of using the mean of the ESC from the 4 limbs. Since the authors chose to use the cut-off of 60 µS to define normal vs abnormal, this dichotomic variable should be used in the logistic regression.
7- The ROC analysis showed interesting results for the diagnostic value of the ESC, in particular regarding DPN, as expected. However, I do not understand the choice of comparing the AUC of ESC with other variables. Since the variables chosen for comparison where the ones used as goal standard for the definition of each condition, ESC would always shown poorer results. Also, it is not clear which cut-off value was used. 60µS? Other?
8- Additionally to the ROC analysis, the authors should clearly present sensitivity, specificity, PPV and NPV for ESC in each condition, in the results section, and not throughout the discussion.
9- In the beginning of the Discussion section, the authors state that “patients with a lower ESC (<60μS) had 2.1-fold increased likelihood of having DPN, 2.4-fold increased likelihood of having DKD, 1.2-fold increased likelihood of having DR, than those with a higher ESC.” However, there is no reference in the results to this analysis. It is not clear where do these results come from.
10- The discussion should be shortened around the main findings of the present article. Results of the study should be clearly stated in the Results section and not scattered though the discussion.
11- The authors state that “it is the first study to report the relationship between sudoscan and DR in T2DM”. However, Camion et al. 2019 (https://doi.org/10.2337/dc18-2202), reported lower feet ESC in patients with severe diabetic retinopathy.
12- The authors suggest that one possible explanation for the lower FESC in male, when compared to females could be “the fact that men are more likely to have bad habits such as smoking”. However, when looking at table 2, one notes that HESC had a higher and stronger correlation with smoking than FESC (-0.134 p = 0.002 vs -0.095 p = 0.031). With the current data, the authors cannot back up this statement.

Reviewer 2 ·

Basic reporting

The study by Lin et. al. evaluates the relationship between parameters of an electrical skin conductance measurement device called SUDOSCAN and microvascular complications in T2DM. It is a well-designed clinical study involving 515 patients with T2DM in China. The methods are appropriately described, results clearly mentioned and conclusions properly drawn. However, there are some concerns that need to be addressed before the manuscript can be considered for publication.

Experimental design

• What was the percentage of males vs. females used for the study? Table-1 describes only males. Please correct accordingly and include this under ‘materials and methods’ section
• How is the value of SUDOSCAN-MDRD calculated? Please include details on how this and other relevant parameters are calculated
• One of the major shortcomings of this manuscript is the absence of an appropriate non-diabetic control group. Are the authors aware of any prior studies done in non-diabetic individuals with measurement of similar parameters? The authors are encouraged to make use of that data and compare it to their own data, with appropriate referencing.
• It would add to the strength of the manuscript if the authors can categorize results under separate subtitles highlighting each finding.
• When citing prior work, please include ‘et. al.’ after author’s name. For example, refer to lines 170, 171, 172. Please change in entire text accordingly.
• Results line 121-122: Please include data values (Mean+/-SEM, P) in text for significant differences in CHD, DPN, DKD and DR

Validity of the findings

• The SUDOSCAN device measures skin conductance based on activity of the sweat glands which can be significantly altered under conditions of physical activity, stress, and blood pressure medications. Were these parameters considered to normalize ESC data?

---

## Round 0.2 · Major Revisions

Although the authors have adequately addressed many concerns of the reviewers, some points in the manuscript (clearly indicated by Reviewer 1) need to be further revised prior to acceptance.

·

Basic reporting

See additional comments

Experimental design

See additional comments

Validity of the findings

See additional comments

Additional comments

The authors submitted a revised version of the original paper, with point by point responses to the questions asked initially.
While there were some changes that improved the manuscript quality, there are still some points that need to be addressed before a publication recommendation.

2- Across the article, the terms Sudoscan and ESC are used interchangeable. This should be avoided. The technique used is the Electrochemical Skin Conductance. The device used for this is called Sudoscan. The name of the device should not be used as a technique. Please clarify this throughout the text

Response: Thanks for your comment. We have checked the whole article and made changes.( line 199, 219, 221, 227 )
While the changes made were adequate, they were clearly insufficient. Some examples of the incorrect use of the term Sudoscan as a technique and not a device:
Line 24 “SUDOSCAN was tested and evaluated with electrochemical skin conductance”
Line 58 “Sudoscan is a method”
Line 200 “The results suggested that sudoscan can be used as a clinical screening tool for diabetic neuropathy”
3- Clarification regarding SUDOSCAN-MDRD is needed. The authors do not explain this score or reference any publication in the Methods section. Why was this score used in the ROC analysis instead of the ESC values?
I have some reservations in using algorithms that are unknown to the authors, since they are proprietary of the firm that sells this specific device and, for that matter, impossible to scrutinize. However, the authors clearly state in the text this particular point (line 89) so readers can be aware of the fact.
6- Another point in the binary logistic regression is the choice of using the mean of the ESC from the 4 limbs. Since the authors chose to use the cut-off of 60 µS to define normal vs abnormal, this dichotomic variable should be used in the logistic regression.
Response: This is a good comment. The main reason we used the average value of ESC instead of the binary classification value is that the registration data will reduce the statistical efficiency compared with the measurement data.
For example, one patient ESC11 and another patient ESC 59. Both are ESC abnormal, their binary classification value are the same, but there are great differences in clinical severity. Therefore, we choose original measurement data.
I have to disagree with the choice to use the original measurement data. The objective of this paper is to assess the potential role of ESC as a biomarker of microvascular diabetic complications. For that matter, the work should focus on cut-off values and their respective diagnostic value.

9- In the beginning of the Discussion section, the authors state that “patients with a lower ESC (<60μS) had 2.1-fold increased likelihood of having DPN, 2.4-fold increased likelihood of having DKD, 1.2-fold increased likelihood of having DR, than those with a higher ESC.” However, there is no reference in the results to this analysis. It is not clear where do these results come from.

Response: This statement is based on table 1. There were significant differences in presence of DPN (88.99% vs 42.86%, P < 0.001), presence of DKD (39.76% vs 16.40%, P < 0.001) and presence of DR (30.58% vs 25.53%, P = 0.012) between both groups. We have realized that this statement was not rigorous. So we changed this sentence to " patients in ESC normal (<60μS) group had 2.1-fold increased likelihood of having DPN, 2.4-fold increased likelihood of having DKD, 1.2-fold increased likelihood of having DR, than those in ESC abnormal (≥60μS) group. " in the text. (line 179-182)
This analysis is methodological incorrect. The fact that in this particular group of patients there was 2.1 more subjects with DPN in the abnormal ESC group, does not mean that subjects with abnormal ESC have 2.1-fold increase in likelihood of having DPN. The same is valid for the other statements.
In order to assess this kind of relationship, the authors should use the binomial logistical regression, with a dichotomic variable of normal/abnormal ESC. The analysis of the odds ratios is what would allow for the authors to make this kind of statements.

7- The ROC analysis showed interesting results for the diagnostic value of the ESC, in particular regarding DPN, as expected. However, I do not understand the choice of comparing the AUC of ESC with other variables. Since the variables chosen for comparison where the ones used as goal standard for the definition of each condition, ESC would always shown poorer results. Also, it is not clear which cut-off value was used. 60µS? Other?

Response: In this study, we compared AUC of ESC with other variables because we want to explored the diagnostic value of ESC. In particular, we wanted to understand the gap between ESC and other gold indicators, and whether it can play an auxiliary or alternative role in clinical diagnosis. Cut-off value was showed in table 4.

The Results sub-section “ROC curve analysis:effective diagnostic value of ESC in diagnosing DPN” (Line 161) repeats data that is given in table 4.
I would advise to cut the text, since the results are well described in table 4 and figure 1.

Reviewer 2 ·

Basic reporting

All the questions have been satisfactorily answered, I have no further comments.

Experimental design

None

Validity of the findings

None

Additional comments

None

---

## Round 0.3 · accepted · Accept

The authors have satisfactorily addressed all the remaining issues raised by the reviewer.

Please consider the additional comments of the reviewer ( in the abstract, line 25, and consistency about the way the word SUDOSCAN is reported throughout the text)

·

Basic reporting

See additional comments

Experimental design

See additional comments

Validity of the findings

See additional comments

Additional comments

The authors incorporated the suggestions given.
It is the opinion of this reviewer that this work now meets publication criteria.

There are only very small details, not pertinent to study design or quality.

Abstract - Line 25, The word SUDOSCAN is repeated, and should be deleted.
The word "Sudoscan" should be written in a consistent fashion throughout the text. It is written either in all caps, with a capital first letter or always in lowercase. Please revise this.